# Relative Lean Body Mass and Waist Circumference for the Identification of Metabolic Syndrome in the Korean General Population

**DOI:** 10.3390/ijerph182413186

**Published:** 2021-12-14

**Authors:** Eunjoo Kwon, Eun-Hee Nah, Suyoung Kim, Seon Cho

**Affiliations:** Health Promotion Research Institute, Korea Association of Health Promotion, 372 Hwagok-ro, Gangseo-Gu, Seoul 07572, Korea; 4ever35@hanmail.net (E.K.); sy.kahp@gmail.com (S.K.); seon@kahp.or.kr (S.C.)

**Keywords:** lean body mass, waist circumference, metabolic abnormality, metabolic syndrome, bioelectrical impedance analysis, optimal cut-offs of relative lean body mass

## Abstract

Lean body mass (LBM) comprises organs and muscle, which are the primary determinants of energy expenditure and regulation of glucose and lipid metabolism. Excessive abdominal fat is associated with metabolic abnormality. Little is known about the relationship between metabolic abnormality and LBM and waist circumference (WC), especially in the Asian general population. The aim of this study was to clarify this relationship. We performed a cross-sectional study with 499,648 subjects who received health check-ups at 16 health promotion centers in 13 Korean cities between January 2018 and October 2019. The subjects were categorized into four groups: (a) High (H)-RLBM (relative lean body mass)/Normal (N)-WC, (b) High-RLBM/Abnormal (A)-WC, (c) Low (L)-RLBM/Normal-WC, and (d) Low-RLBM/Abnormal-WC. RLBM was calculated using fat mass data that were estimated via bioelectrical impedance analysis. L-RLBM/A-WC was significantly associated with metabolically unhealthy status (OR: 4.40, 95% CI: 4.326–4.475) compared to H-RLBM/N-WC. L-RLBM/N-WC (OR: 2.170, 95% CI: 2.122–2.218) and H-RLBM/A-WC (OR: 2.713, 95% CI: 2.659–2.769) were also significantly related to metabolic unhealthy status. The cut-offs of RLBM for predicting metabolic syndrome (MetS) were 74.9 in males and 66.4 in females (*p* < 0.001). L-RLBM and A-WC are associated with metabolic abnormality in the Korean general population. RLBM is an anthropometric index that can be used to predict MetS in primary health care.

## 1. Introduction

Obesity is associated with metabolic abnormalities, such as insulin resistance, prediabetes, nonalcoholic fatty liver disease, and metabolic syndrome (MetS). Obesity is often accompanied by multiple cardiovascular risk factors [1,2]. Almost 603.7 million adults in the world are obese, and the prevalence of obesity, which has doubled in 73 countries since 1980, has increased in almost every country continuously [3].

Obesity is diagnosed in adults most commonly using the body mass index (BMI), as recommended by the World Health Organization (WHO) [4]. BMI is calculated from the weight-adjusted height, and it is known as a good measure of general adiposity. However, BMI cannot measure body composition accurately, because it does not distinguish lean body mass (LBM) from fat mass [5,6,7]. Moreover, the association between obesity and metabolic complications is partly dependent on the pattern of body fat distribution [8]. Because of the limitations of BMI, the use of other anthropometric measures, such as the waist circumference (WC) or the waist-to-hip ratio, has been suggested to better estimate obesity-related health risks [9]. The WC is important in clinical practice as a screening tool for abdominal fat distribution and abdominal obesity. Previous studies have shown that WC, either singly or in combination with BMI, may have a more powerful relationship with some health outcomes than BMI alone [9,10,11].

LBM comprises organs and muscle. LBM plays a critical role in whole-body glucose disposal and energy expenditure [12]. Fat mass functions as a site of energy storage and an endocrine regulator of energy metabolism [13]. Therefore, while too much fat mass has been shown to be detrimental for health, LBM may be beneficial for health, as skeletal muscle accounts for most of the LBM [14]. Previous studies have suggested that a low LBM is associated with a higher incidence of cardiometabolic events and diabetes [15,16,17].

Although some studies have been conducted on the association between LBM and cardiometabolic risk factors [12,15,16,18,19,20], there are very few studies that have investigated the concurrent contribution of LBM and WC to metabolic abnormalities in the general population.

The aims of this study were to assess the association between relative lean body mass (RLBM) and metabolic abnormalities and to determine the ability of RLBM to identify the presence of metabolic syndrome in the Korean general population.

## 2. Materials and Methods

### 2.1. Study Subjects

This cross-sectional retrospective study consecutively selected subjects who went to health check-ups at 16 health promotion centers in 13 Korean cities between January 2018 and October 2019. Study subjects with missing results on measurements of anthropometric and metabolic parameters were excluded. Finally, a total of 499,648 subjects were analyzed in this study (Figure 1). The study protocol was reviewed and approved by the institutional review board of the Korean Association of Health Promotion (130750-202009-HR-017).

### 2.2. Measurement of Lean Body Mass

Body weight in kg and height in cm were measured to the nearest 0.1 units using an automatic portable stadiometer (BSM 370, Biospace Inc., Seoul, Korea), while the study subjects were dressed in light clothing without shoes. BMI was calculated as the weight (kg) divided by height squared (m^2^). WC was taken using soft tape at the point between the upper iliac crest and the lowest rib after normal expiration. Abnormal WC was defined as WC ≥90 cm in males and ≥85 cm in females [21].

Fat mass was estimated via bioelectrical impedance analysis (BIA) measurement using an eight-point tactile electrode multifrequency BIA device, according to the manufacturer’s instructions (InBody 770; Biospace Inc., Seoul, Korea), after overnight fasting for at least 10 h. BIA is a simple, invasive, and useful field method for assessing body composition, especially muscle mass, on a large scale [22,23]. However, BIA is difficult to measure accurately a specific body composition in which the degree of hydration is inconstant [22,24]. A consistent environment and location are required in the assessment of body composition using BIA [22]. The assessment of body composition by BIA relies on a calibration equation developed using a reference such as whole-body dual-energy X-ray absorptiometry (DEXA), computed tomography (CT), and magnetic resonance imaging [22,25]. BIA devices for qualitative approach have been developed, and these allow raw data to be inserted into regression equations [22]. The InBody 770 system measured body composition across the whole body and 5 segments (right arm, left arm, trunk, right leg, and left leg). BIA measurement consisted of two combinations: bioelectrical impedance (z) at 6 different frequencies (1, 5, 50, 250, 500, 1000 Khz) for impedance and reactance (Xc) at 3 different frequency (5, 50, 250 kHz) [26]. The total body impedance value was calculated as the sum of the segmental impedance values. Resistance (R) was calculated according to the BIA generator formula V = ρ × height^2^/resistance [24,27]. Reactance (R) was calculated using trigonometric formula Z^2^ = R^2^ + XC^2^. The bioelectrical phase angle (φ) was calculated as the arctangent of Xc/R × 180°/π [22,24,27]. For the body composition assessment, participants stood barefoot on the platform of the InBody 770 system in an upright position, with their feet centered on the electrodes. Then, they grasped the hand electrodes with their arms held wide enough. In this large-scale study, estimates of body composition parameters calculated by quantitative analysis were collected. LBM was calculated as the difference between total body weight and body fat weight. Systolic/diastolic blood pressure (SBP/DBP) was measured on the right arm after resting for ≥5 min using a digital sphygmomanometer (TM-2657P, A&D Co. Ltd., Tokyo, Japan) with an appropriate cuff size in a sitting position. RLBM was defined as lean body mass divided by body weight in percentage.

### 2.3. Laboratory Measurements

Overnight 10–12 h fasting blood samples were provided for measurement of triglycerides (TG), high-density lipoprotein cholesterol (HDL-C), and fasting blood sugar (FBS). TG and HDL-C were measured using enzymatic methods, and serum glucose was measured using hexokinase methods by an autoanalyzer (Hitachi 7600, Hitachi high-technologies CO., Tokyo, Japan).

### 2.4. Grouping of Study Subjects Using the Combination of LBM and WC

Subjects in the lowest quartile of RLBM were considered Low-RLBM, while those in the 2nd to 4th quartiles of RLBM were considered High-RLBM. Abnormal WC was defined as WC ≥90 cm and ≥85 cm in males and females, respectively. The study subjects were categorized into four groups based on WC and RLBM: (a) High (H)-RLBM/normal (N)-WC, (b) High-RLBM/abnormal (A)-WC, (c) Low (L)-RLBM/N-WC, and (d) Low-RLBM/A-WC.

### 2.5. Definition of Metabolically Unhealthy Status

MetS diagnosis was based on the five components of the Third Report of the Cholesterol Education Program Adult Treatment PanelⅢ (NCEP-ATPⅢ) criteria for MetS [28] and the current standard for abdominal obesity in Korean males and females [21]. The components of MetS include TG >150 mg/dL, HDL-C <40 mg/dL in males and <50 mg/dL in females, FBS >100 mg/dL, SBP/DBP >130/85 mmHg, and WC ≥90 cm in males ≥85 cm in females, respectively.

“Metabolically unhealthy” was defined as having ≥2 of the MetS components, except abdominal obesity.

### 2.6. Statistical Analysis

Categorical variables are expressed as frequency with percentage, and continuous variables are presented as mean ± standard deviation (SD). A Chi-square test was used to assess differences in categorical variables among groups. A one-way ANOVA was used to identify differences among groups for continuous variables. The logistic regression model was applied to estimate the association of metabolic abnormalities with groups of subjects compared to the reference type (H-RLBM/N-WC). The performance of RLBM for predicting MetS was assessed using area under the receiver operating characteristic curves (AUROC). The AUROC and the optimal cut-offs for MetS prediction of RLBM were determined by the Youden index. A value of *p* < 0.05 was considered significant. Data management and analysis were performed using SPSS software version 21 (SPSS Inc., Chicago, IL, USA).

## 3. Results

### 3.1. Characteristics of the Study Subjects According to Categories of RLBM

The characteristics of the study subjects and categories of RLBM are presented in Table 1. The mean age of the study subjects was 48.0 ± 12.5 years (range: 19–97 years). The mean ± SD of age, BMI, and WC was 47.7 ± 12.2 years (range: 19–97 years), 25.1 ± 6.1 kg/m^2^, and 86.6 ± 8.4 cm in males and 48.3 ± 12.9 years (range: 19–92 years), 23.2 ± 3.4 kg/m^2^, and 76.9 ± 9.4 cm in females, respectively. A total of 35.7% of the male subjects and 20.4% of the female subjects were metabolically unhealthy. Males and females in Q1 of RLBM were more likely to be metabolically unhealthy; were more likely to have hypertension, type 2 diabetes, and MetS; had higher levels for BMI, WC, SBP, DBP, FBS, and TG; and had lower levels of HDL-C (*p* < 0.001).

### 3.2. Prevalence of Metabolic Abnormalities in Groups of Subjects

Table 2 presents the prevalence of metabolic abnormalities among the four groups, according to the combination of RLBM and WC. The prevalence rates in H-RLBM/N-WC, L-RLBM/N-WC, H-RLBM/A-WC, and L-RLBM/A-WC were 60.8%, 6.7%, 14.2%, and 18.3% in males and 69.3%, 12.0%, 5.7%, and 13.0% in females, respectively. The L-RLBM/A-WC group had the highest prevalence of metabolic abnormalities compared to H-RLBM/N-WC in both sexes (*p* < 0.001).

### 3.3. Association of Metabolic Abnormalities with Groups of Subjects

Table 3 presents the associations of metabolic abnormalities with the different groups of subjects. L-RLBM/A-WC was significantly associated with “metabolically unhealthy” status (odds ratio, OR: 4.40, 95% CI: 4.326–4.475) after adjusting for age and sex compared with H-RLBM/N-WC (*p* < 0.001). L-RLBM/N-WC (OR: 2.170, 95% CI: 2.122–2.218) and H-RLBM/A-WC (OR: 2.713, 95% CI: 2.659–2.769) were significantly correlated with “metabolically unhealthy” status compared with H-RLBM/N-WC.

### 3.4. Determination of the Optimal Cut-Offs of RLBM for Predicting Metabolic Syndrome

Figure 2 shows the ROC curves for assessing the performance of RLBM in the prediction of MetS in both sexes. The AUCs of RLBM were 0.755 (95% CI: 0.753–0.758) in males and 0.797 (95% CI: 0.794–0.799) in females, respectively (*p* < 0.001). The cut-offs of RLBM for predicting metabolic syndrome were 74.9 in males and 66.4 in females with the most optimal sensitivity and specificity, respectively.

## 4. Discussion

This study found that low RLBM and abnormal WC (abdominal obesity) were associated with metabolic abnormalities, especially in those with both low RLBM and abdominal obesity, who had the highest risk of metabolic abnormality.

MetS is a cluster of risk factors linked to abdominal obesity and insulin resistance, which has been associated with the subsequent development of type 2 diabetes and cardiovascular disease [29]. Insulin plays a key role in energy balance and glucose homeostasis. Insulin works on the insulin receptor, a membrane-bound tyrosine kinase [30]. Skeletal muscle is the major tissue in glucose and energy metabolism because insulin receptors are located in the liver, skeletal muscle, and adipose tissue. High skeletal muscle mass might contribute to a stable control over glucose levels due to insulin-induced glucose uptake occurring in skeletal muscle [31,32]. Therefore, reduced skeletal muscle mass may lead to an increase in insulin resistance and metabolic abnormalities [33].

In this study, RLBM was associated with metabolic abnormalities. The low RLBM groups had a higher prevalence of metabolic abnormalities than the high RLBM groups. The RLBM used in this study was the LBM adjusted by body weight. This can be easily used instead of skeletal muscle mass in primary health care settings. Thus, low RLBM may have affected insulin resistance and metabolic disturbance in this study. This finding is similar to a previous study, which suggested that weight-adjusted LBM measured by dual-frequency BIA and CT is protective against obesity-related insulin resistance and metabolic abnormalities in Japanese adults without diabetes [12]. In a Korean cohort study, an increase in relative skeletal muscle mass measured by direct segmentation in standing position at multifrequency BIA using the same technologies as used in this study had an inverse association with the development of MetS over time [19]. Additionally, relative skeletal muscle, such as the skeletal muscle to visceral fat ratio (SVR) measured through direct segmentation in standing position at multifrequency BIA using the same technologies as our study, was found to be negatively associated with an increased risk of exacerbating MetS [20,34].

In addition, our study showed that A-WC had a larger effect on metabolic abnormalities than RLBM. There are few studies confirming the association between muscle mass, abdominal obesity, and MetS. In one study, the accumulation of abdominal fat was related to insulin resistance and MetS, independent of muscle mass, in Korean adults [35]. Abdominal (visceral) fat deposit is a major source of free fatty acids and is associated with a change in the normal balance of adipokines, which leads to a proinflammatory state [36,37]. An increase in total body fat and abdominal adiposity can be accompanied by overaccumulation of various lipids in skeletal muscle cells [38]. Intramyocellular lipids in particular may play a leading role in modulating insulin sensitivity [38,39]. An increase in intramyocellular lipids has been linked to elevation of the insulin resistance of skeletal muscles [39,40]. Thus, excessive abdominal fat may weaken the protective effects of skeletal muscle against MetS [35,39].

There are many studies that have predicted the incidence of MetS using LBM [12,19,20,35]. Some of these studies confirmed the association between relative skeletal muscle mass and development of MetS using multifrequency BIA devices [19,20]. Studies among Asian populations have validated the relationship between insulin resistance and metabolic abnormalities using BIA devices, CT scanning, or DEXA [12,35]. Unlike these studies, we attempted to establish cut-off values for RLBM in predicting MetS, based on a large sample, using BIA devices that are relatively easy to access in primary health care.

The current study has some notable strengths. To the best of our knowledge, this is the first study to investigate the association between metabolically unhealthy status and groups according to the combination of LBM and WC to determine cut-off values for RLBM in predicting MetS in the Korean general population. Second, our study involves a large and nationwide sample from 16 health promotion centers in 13 cities across the country. Participants were recruited from the national health screening program, which suggests that this study population was representative of the Korean general population.

On the other hand, our study has some limitations. First, our results cannot be regarded as a direct causal relationship among RLBM, WC, and metabolic abnormalities due to the cross-sectional design of the study. Second, we assessed LBM using BIA, which is not considered the gold-standard technique. However, this method is simple, inexpensive, and has good sensitivity (85%) and specificity (100%) compared to DEXA [41]. The BIA method has also been found to be accurate for the assessment of fat-free mass (FFM), known as LBM, in epidemiological studies [42]. The body composition parameters, such as LBM, are dependent on the technologies of the measuring instrument. Therefore, it is necessary to be careful to compare with the results of other studies that estimate body composition with different technologies. Third, we used data for LBM that included the weight of organs, skin, bones, body water, and muscles. Thus, different results may have been obtained had we used data on skeletal muscle mass, such as appendicular muscle. Finally, our study did not include socioeconomic status and lifestyle factors such as nutrition, exercise, smoking, and alcohol use. The inclusion of other variables will be considered in a future study.

## 5. Conclusions

In conclusion, L-RLBM and A-WC are associated with metabolic abnormalities in the general population. Abdominal obesity has a greater effect on metabolic abnormalities than RLBM. Additionally, it is possible to predict MetS using RLBM, which is an easily accessible anthropometric index in primary health care. Thus, RLBM can be used as a surrogate marker for identifying MetS in the Korean general population.

## Figures and Tables

**Figure 1 ijerph-18-13186-f001:**
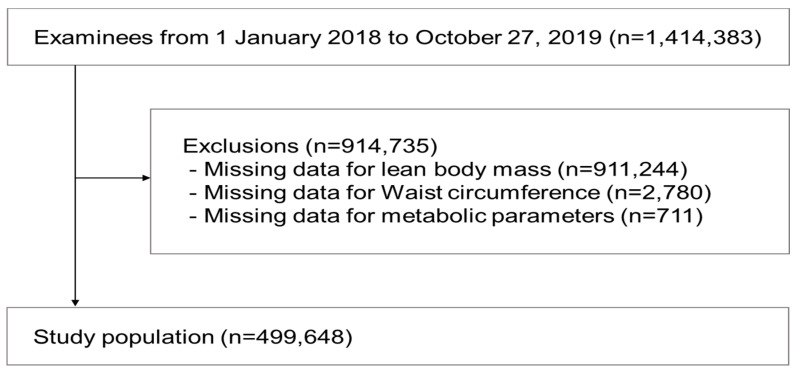
Study flow chart.

**Figure 2 ijerph-18-13186-f002:**
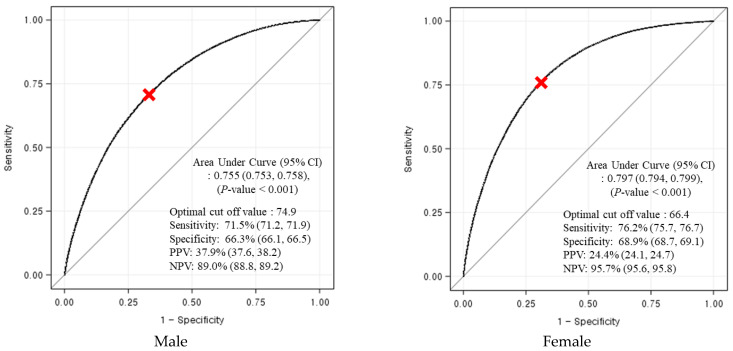
Receiver operating characteristic curve and cut-off values of relative lean body mass in predicting metabolic syndrome. Abbreviations. CI, confidence interval; NPV, negative predictive value; PPV, positive predictive value.

**Table 1 ijerph-18-13186-t001:** Characteristics of subjects by RLBM quartile.

	Total		Q1		Q2		Q3		Q4		*p*-Value
Overall											
RLBM (%), Range			13.96–67.77		67.78–72.77		72.78–77.06		77.07–98.86		
Number, N	N = 499,648		N = 124,912		N = 124,929		N = 124,895		N = 124,912		
Males											
RLBM (%), Range			13.96–72.48		72.49–75.89		75.90–79.33		79.34–98.86		
Number, N	263,735		65,941		65,927		65,947		65,920		<0.001
Age (year)	47.7	(12.2)	47.7	(12.8)	48.6	(11.9)	48.1	(11.7)	46.4	(12.5)	<0.001
Hypertension, N (%)	97,368	(36.9)	34,160	(51.8)	26,093	(39.6)	21,564	(32.7)	15,551	(23.6)	<0.001
Type 2 diabetes, N (%)	29,710	(11.3)	10,872	(16.5)	7963	(12.1)	6294	(9.5)	4581	(6.9)	<0.001
Metabolically unhealthy, N (%)	94,024	(35.7)	35,013	(53.1)	26,591	(40.3)	20,722	(31.4)	11,698	(17.7)	<0.001
Metabolic syndrome, N (%)	58,966	(22.4)	29,976	(45.5)	16,439	(24.9)	9134	(13.9)	3417	(5.2)	<0.001
BMI (kg/m^2^)	25.1	(6.1)	28.2	(3.1)	25.7	(10.6)	24.2	(2.1)	22.3	(2.2)	<0.001
WC (cm)	86.6	(8.4)	94.6	(7.7)	88.2	(5.5)	84.5	(5.4)	78.9	(6.1)	<0.001
Systolic BP (mmHg)	120.3	(13.1)	124.4	(13.2)	120.9	(12.9)	119.2	(12.6)	116.8	(12.3)	<0.001
Diastolic BP (mmHg)	76.6	(9.4)	79.5	(9.6)	77.2	(9.2)	75.9	(9.0)	73.9	(8.9)	<0.001
FBS (mg/dL)	100.3	(22.4)	104.8	(25.3)	101.5	(22.2)	99.1	(20.6)	96.0	(20.4)	<0.001
HDL-C (mg/dL)	50.5	(11.6)	47.1	(10.0)	48.8	(10.6)	50.7	(11.3)	55.3	(12.8)	<0.001
TG (mg/dL)	145.8	(106.1)	177.4	(118.1)	158.8	(109.5)	140.1	(99.4)	106.9	(80.6)	<0.001
Females											
RLBM (%), Range			15.17–64.52		64.53–68.46		68.47–72.58		72.59–97.85		
Number, N	235,913		59,017		58,944		58,987		58,965		<0.001
Age (year)	48.3	(12.9)	52.4	(13.1)	50.5	(12.5)	47.5	(12.1)	42.9	(11.7)	<0.001
Hypertension, N (%)	55,714	(23.6)	24,129	(40.9)	15,538	(26.4)	10,214	(17.3)	5833	(9.9)	<0.001
Type 2 diabetes, N (%)	15,917	(6.7)	7306	(12.4)	4431	(7.5)	2740	(4.6)	1440	(2.4)	<0.001
Metabolically unhealthy, N (%)	48,066	(20.4)	22,312	(37.8)	14,051	(23.8)	8274	(14.0)	3429	(5.8)	<0.001
Metabolic syndrome, N (%)	27,430	(11.6)	17,016	(28.8)	6877	(11.7)	2732	(4.6)	805	(1.4)	<0.001
BMI (kg/m^2^)	23.2	(3.4)	26.9	(3.4)	23.8	(2.1)	22.0	(1.7)	20.0	(1.6)	<0.001
WC (cm)	76.9	(9.4)	85.6	(8.5)	78.7	(7.1)	74.3	(5.8)	69.0	(6.6)	<0.001
Systolic BP (mmHg)	114.7	(14.4)	121.5	(14.4)	116.2	(13.8)	112.4	(13.4)	108.6	(12.5)	<0.001
Diastolic BP (mmHg)	71.8	(9.2)	75.5	(9.3)	72.6	(8.9)	70.5	(8.6)	68.5	(8.2)	<0.001
FBS (mg/dL)	94.4	(17.3)	99.9	(20.9)	95.7	(17.4)	92.5	(15.1)	89.3	(13.0)	<0.001
HDL-C (mg/dL)	60.1	(13.5)	55.7	(12.2)	58.2	(12.8)	61.0	(13.2)	65.5	(13.6)	<0.001
TG (mg/dL)	96.4	(63.6)	119.3	(72.1)	105.0	(67.5)	89.5	(57.6)	71.8	(43.2)	<0.001

Note: Data are presented as frequency (percentage) or mean (standard deviation) values. The *p*-values are from ANOVA for continuous variables and Chi-square tests for categorical variables. Relative lean body mass was calculated as the lean body mass (kg) divided by body weight (kg) in percentage. Hypertension, BP ≥ 130/85 mmHg, or use of antihypertensive medication, including diuretics, beta-blockers, ACE inhibitors, angiotensin II receptor blockers, calcium channel blockers, alpha blockers, and alpha-2 receptor agonists; type 2 diabetes, fasting glucose ≥ 126 mg/dL, or HbA1c level ≥ 6.5%, or use of antidiabetic agents, including alpha-glucosidase inhibitors, amylin analogs, sulfonylurea, nonsulfonylureas, SGLT-2 inhibitors, thiazolidinediones, and insulin; metabolic syndrome, three or more NCEP-ATPⅢ criteria; metabolically unhealthy, two or more NCEP-ATPⅢ criteria except abdominal obesity. NCEP-ATPⅢ criteria: WC ≥ 90 cm in males, ≥85 cm in females (Korean abdominal obesity criteria); BP ≥ 130/85 smmHg or antihypertension medication use; FBS ≥ 100 mg/dL; TG ≥ 150 mg/dL; HDL-C < 40 mg/dL in males, <50 mg/dL in females. Abbreviations: Q, quartile; BMI, body mass index; WC, waist circumference; BP, blood pressure; FBS, fasting blood sugar; HDL-C, high-density lipoprotein cholesterol; TG, triglycerides.

**Table 2 ijerph-18-13186-t002:** The prevalence of metabolic abnormalities in groups of subjects.

	Total		H-RLBM/N-WC		L-RLBM/N-WC		H-RLBM/A-WC		L-RLBM/A-WC		*p*-Value
Overall	499,648	(100.0)	323,960	(64.8)	46,039	(9.2)	50,730	(10.2)	78,919	(15.8)	
Males	263,735	(100.0)	160,394	(60.8)	17,662	(6.7)	37,400	(14.2)	48,279	(18.3)	
BP ≥ 130/85 mmHg	80,619	(30.6)	38,458	(24.0)	6604	(37.4)	13,618	(36.4)	21,939	(45.4)	<0.001
FBS ≥ 100 mg/dL	95,286	(36.1)	48,150	(30.0)	7278	(41.2)	16,358	(43.7)	23,500	(48.7)	<0.001
TG ≥ 150 mg/dL	92,662	(35.1)	42,879	(26.7)	7457	(42.2)	17,032	(45.5)	25,294	(52.4)	<0.001
HDL-C < 40 mg/dL	41,108	(15.6)	18,564	(11.6)	3049	(17.3)	7921	(21.2)	11,574	(24.0)	<0.001
Metabolically unhealthy	94,024	(35.7)	41,459	(25.8)	7728	(43.8)	17,552	(46.9)	27,285	(56.5)	<0.001
Females	235,913	(100.0)	163,566	(69.3)	28,377	(12.0)	13,330	(5.7)	30,640	(13.0)	
BP ≥ 130/85 mmHg	43,709	(18.5)	20,571	(12.6)	7198	(25.4)	3893	(29.2)	12,047	(39.3)	<0.001
FBS ≥ 100 mg/dL	52,269	(22.2)	26,071	(15.9)	7976	(28.1)	4705	(35.3)	13,517	(44.1)	<0.001
TG ≥ 150 mg/dL	32,416	(13.7)	15,483	(9.5)	5140	(18.1)	3210	(24.1)	8583	(28.0)	<0.001
HDL-C < 50 mg/dL	51,003	(21.6)	27,366	(16.7)	7634	(26.9)	4471	(33.5)	11,532	(37.6)	<0.001
Metabolically unhealthy	48,066	(20.4)	20,928	(12.8)	7849	(27.7)	4826	(36.2)	14,463	(47.2)	<0.001

Note: Data are presented as the frequency (percentage). *p*-values are from Chi-square tests. Low-RLBM, relative LBM ≤ the first quartile (Q1); high-LBM, from the second quartile (Q2) to the fourth quartile (Q4); normal-WC, WC < 90 cm in males, <85 cm in females; abnormal WC, WC ≥ 90 cm in males, ≥85 cm in females; metabolic unhealthy, two or more NCEP-ATPⅢ criteria except abdominal obesity. Relative lean body mass quartile level. Males: Quartile 1, 13.96–72.48%; Quartile 2, 72.49–75.89%; Quartile 3, 75.90–79.33%; Quartile 4, 79.34–98.86%. Females: Quartile 1, 15.17–64.52%; Quartile 2, 64.53–68.46%; Quartile 3, 68.47–72.58%; Quartile 4, 72.59–97.85%; NCEP-ATPⅢ criteria: WC ≥ 90 cm in males, ≥85 cm in females (Korean abdominal obesity criteria); BP ≥ 130/85 mmHg or antihypertension medication use; FBS ≥ 100 mg/dL; TG ≥ 150 mg/dL; HDL-C < 40 mg/dL in males, <50 mg/dL in females. Abbreviations: H, high; RLBM, relative lean body mass; N, normal; WC, waist circumference; L, low; A, abnormal; BP, blood pressure; FBS, fasting blood sugar; TG, triglycerides; HDL-C, high-density lipoprotein cholesterol.

**Table 3 ijerph-18-13186-t003:** The odds ratio of components of metabolically unhealthy status in subjects according to relative lean body mass and waist circumference.

Metabolic Abnormality	H-RLBM/N-WC	L-RLBM/N-WC			H-RLBM/A-WC			L-RLBM/A-WC		
		OR	(95%CI)	*p*-Value	OR	(95%CI)	*p*-Value	OR	(95%CI)	*p*-Value
High BP	Ref	1.891	(1.849–1.934)	<0.001	1.936	(1.896–1.977)	<0.001	3.110	(3.057–3.163)	<0.001
High FBS	Ref	1.598	(1.563–1.634)	<0.001	1.928	(1.889–1.967)	<0.001	2.714	(2.669–2.761)	<0.001
High TG	Ref	1.989	(1.943–2.036)	<0.001	2.411	(2.362–2.461)	<0.001	3.195	(3.140–3.250)	<0.001
Low HDL-C	Ref	1.638	(1.599–1.678)	<0.001	2.117	(2.068–2.166)	<0.001	2.560	(2.513–2.609)	<0.001
Metabolically unhealthy	Ref	2.170	(2.122–2.218)	<0.001	2.713	(2.659–2.769)	<0.001	4.400	(4.326–4.475)	<0.001

Odds ratio was adjusted by age and sex. Abbreviations: H, high; RLBM, relative lean body mass; N, normal; WC, waist circumference; L, low; A, abnormal; OR, odds ratio; CI, confidence interval; BP, blood pressure; FBS, fasting blood sugar; TG, triglycerides; HDL-C, high-density lipoprotein cholesterol.

## Data Availability

No data are available.

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
