# Peer review of "Relative Lean Body Mass and Waist Circumference for the Identification of Metabolic Syndrome in the Korean General Population"

_ijerph, 2021, doi:10.3390/ijerph182413186_

Round 1
Reviewer 1 Report
Kwon et al in the research article, "Relative Lean Body Mass and Waist Circumference for the Identification of Metabolic Syndrome in the Korean General Population", report a relationship between the metabolic abnormality and lean body mass and waist circumference. The study is interesting but I have some remarks:
- "An abnormal WC was defined as WC ≥90 cm in males, ≥85 cm in females", which definition has been followed to define these values?
- In section 2.5 the authors define the components of MetS where they say that "WC ≥ 90 cm in males ≥ 80 cm in females, respectively". This is not the same as above mentioned definition for WC. How do the authors address this anomaly?
- The authors should mention if the patients were on any cardiovascular medication. Include a table where you clarify and classify which medication or class of drugs the patients were using, for example, ASA Beta-blocker statins, ACE/ARB, etc. Merely mentioning anti-diabetic or anti-hypertension medication does not suffice.
Author Response
Answers to Reviewer’s Comments (Reviewer 1)
Major Comments
- "An abnormal WC was defined as WC ≥90 cm in males, ≥85 cm in females", which definition has been followed to define these values?
Response)
Authors added a reference to an abnormal WC in 2.2. Measurement of Lean Body Mass (page 2, Line 33-34) as follows: An abnormal WC was defined as WC≥90cm in males, ≥85cm in females [21]. And authors added No. 21 reference in Reference lists (page 10, Line 17-18) as follows: 21. Yoon, Y.S.; Oh, S.W. Optimal waist circumference cutoff values for the diagnosis of abdominal obesity in Korean adults. Endocrinol Metab (Seoul) 2014, 29(4), 418-426, doi: 10.3803/EnM.2014.29.4.418.
- In section 2.5 the authors define the components of MetS where they say that "WC ≥ 90 cm in males ≥ 80 cm in females, respectively". This is not the same as above mentioned definition for WC. How do the authors address this anomaly?
Response)
Authors mistyped “≥80cm in females, respectively” and revised to “≥ 85 cm in females” of the WC criteria in 2.5. Definition of Metabolically Unhealthy Status (page 3, Line 46-49) as follows: MetS diagnosis was based on the five components of the Third Report of the Cholesterol Education Program Adult Treatment PanelⅢ (NCEP-ATPⅢ) criteria for MetS [22] and the current standard for abdominal obesity in Korean male and female [21]. The components of MetS include TG > 150 mg/dL, HDL-C < 40 mg/dL in males and <50 mg/dL in females, FBS > 100 mg/dL, SBP/DBP > 130/85 mmHg, and WC ≥ 90 cm in males ≥ 85 cm in females, respectively.
And authors corrected the definition for abdominal obesity in Table 1 and Table 2 (page , line 20, page 6, line 17) as follows: WC ≥ 90 cm in males, ≥ 85 cm in females (Korean abdominal obesity criteria).
- The authors should mention if the patients were on any cardiovascular medication. Include a table where you clarify and classify which medication or class of drugs the patients were using, for example, ASA Beta-blocker statins, ACE/ARB, etc. Merely mentioning anti-diabetic or anti-hypertension medication does not suffice.
Response)
Authors provided additional anti-hypertensive or anti-diabetic medications the participants were using in Table 1 (page 5, line 15~18) follow as: Hypertension, BP≥130/85 mmHg, or use of anti-hypertensive medication including diuretics, beta-blockers, ACE inhibitors, angiotensin II receptor blockers, calcium channel blockers, alpha blockers, and alpha-2 receptor agonists.; type 2 diabetes, fasting glucose ≥ 126 mg/dL, or HbA1c level ≥6.5%, or use of antidiabetic agents including alpha-glucosidase inhibitors, amylin analogs, sulfonylurea, non-sulfonylureas, SGLT-2 inhibitors, thiazolidinediones, and insulin.
Thank you very much!
Reviewer 2 Report
This research paper deals with an important public health problem; the value of lean body mass in prediction of MetS is tested in a very large cohort. The text is well written, methodology and results are clear and concise. I have no major remarkas, just a minor comment.
Please check the first-time appearing abbreviations throughout the text, for example in page 2, RLBM is not explained, but later in the text.
Author Response
Answers to Reviewer’s Comments (Reviewer 2)
Minor Comment:
Please check the first-time appearing abbreviations throughout the text, for example in page 2, RLBM is not explained, but later in the text.
Response)
Authors corrected the first-time appearing abbreviations throughout the manuscripts as follows:
- Introduction (page 1, line 31): metabolic syndrome (MetS)
- Introduction (page 1, line 38-39): lean body mass (LBM)
- Introduction (page 2, line 13-14): relative lean body mass (RLBM)
- 2 Measurement of Lean body Mass (page 3, line 9-10): Dual Energy X-ray Absorptiometry (DEXA), computed tomography (CT)
Thank you very much!

Reviewer 3 Report
The authors addressed a really interesting topic!
They investigated the relationship between metabolic abnormality and lean body mass and waist circumference in the Asian general population.
Findings showed that relative lean body mass is an anthropometric index that can be used to predict Metabolic Syndrome in primary health care.
In my opinion, there needs to be some revisions to make the manuscript worthy of publication in IJERPH.
Abstract
To optimize the search for the article through search engines, you must always insert keywords that are not already present in the title. I recommend you replace them.
Materials and Methods
Measurement of Lean Body Mass
There is a basic need to describe the technical characteristics of InBody, Japan device. What is the calibration method to ensure validity (accuracy and precision) of the bioimpedance measurements? What is the technical error of measurement in vivo? Provide readers with a concise description of what this BIA device measures. In particular, what are the measurements detected by this tool? Do they directly measure the raw bioimpedance parameters (e.g., R, Xc and phase angle)? Again, what equation was used to estimate body composition parameters? Is it an equation developed using the Inbody device or an instrument that works with similar characteristics (frequency and technologies)?
Discussion
it is important to consider that body composition parameters are dependent instrument and that the instrumental sensitivities are different. Therefore, no comparisons can be made between studies that estimate body composition with different technologies (e.g., foot-to-hand- or direct segmental in standing position) or sampling frequencies. It is also crucial to highlight the difference between a qualitative and a quantitative BIA analysis. Please refer to: Nutrients 2021, 13(5), 1620; https://doi.org/10.3390/nu13051620
Author Response
Answers to Reviewer’s Comments (Reviewer 3)
Major Comments
- Abstract
To optimize the search for the article through search engines, you must always insert keywords that are not already present in the title. I recommend you replace them.
Response)
Authors added keywords in Abstract (page 1, line 26-27) as follow: lean body mass; waist circumference; metabolic abnormality; metabolic syndrome; bioelectrical impedance analysis, optimal cut-offs of relative lean body mass
- Materials and Methods
Measurement of Lean Body Mass
There is a basic need to describe the technical characteristics of InBody, Japan device. What is the calibration method to ensure validity (accuracy and precision) of the bioimpedance measurements? What is the technical error of measurement in vivo? Provide readers with a concise description of what this BIA device measures. In particular, what are the measurements detected by this tool? Do they directly measure the raw bioimpedance parameters (e.g., R, Xc and phase angle)? Again, what equation was used to estimate body composition parameters? Is it an equation developed using the Inbody device or an instrument that works with similar characteristics (frequency and technologies)?
Response)
Authors revised the technical characteristics of InBody in in 2.2. Measurement of Lean Body Mass (page 3, Line 4-24) as follows: BIA is simple, invasive, and useful field method for assessing body composition, especially muscle mass on a large scale [22,23]. But BIA is difficult to measure actually a specific body composition in which the degree of hydration is inconstant [22,24]. And a consistent environment and location is required in the assessment of body composition using BIA [22]. The assessment of body composition by BIA relies on a calibration equation developed using a reference such whole-body Dual Energy X-ray Absorptiometry (DEXA), computed tomography (CT), and magnetic resonance imaging [22,25]. Recently, BIA devices for qualitative approach developed, and theses provide raw data to be inserted into regression equations [22]. The InBody 770 system measured body composition across the whole body and 5 segments (right arm, left arm, trunk, right leg, and left leg). BIA measurement consisted two combinations: bioelectrical impedance (z) at 6 different frequencies (1, 5, 50, 250, 500, 1000Khz) for impedance and reactance (Xc) at 3 different frequency (5, 50, 250kHz) [26]. The total body impedance value was calculated as the sum of the segmental impedance values. The resistance (R) was calculated according to the BIA generator formula V=ρ ˟ height2 / resistance [24,27]. And the reactance (R) was calculated using trigonometric formula Z2 = R2 + XC2. The bioelectrical phase angel (φ) was calculated as the arctangent of Xc / R ˟ 180°/π [22,24,27]. For the body composition assessment, participants stood barefoot on the platform of the InBody 770 in an upright position, the feet centered on the electrodes. Then they grasped the hand electrodes with arms being held wide enough. In this large-scale study, the estimation of body composition parameters which calculated by quantitative analysis were collected.
And authors added 6 references (reference no. 22 ~ 27.) in Reference (page 10, Line 19-32).
- Discussion
it is important to consider that body composition parameters are dependent instrument and that the instrumental sensitivities are different. Therefore, no comparisons can be made between studies that estimate body composition with different technologies (e.g., foot-to-hand- or direct segmental in standing position) or sampling frequencies. It is also crucial to highlight the difference between a qualitative and a quantitative BIA analysis. Please refer to: Nutrients 2021, 13(5), 1620; https://doi.org/10.3390/nu13051620
Response)
Authors revised the comparisons between studies in 4. Discussion (page 7, Line 29~30, page 8, Line 1~9) as follows: This finding is similar to a previous study which suggested that weight-adjusted LBM measured by dual-frequency BIA and CT is protective against obesity-related insulin resistance and metabolic abnormalities in Japanese adults without diabetes [12]. In a Korean cohort study, an increase in relative skeletal muscle mass measured by direct segmental in standing position at multi-frequency BIA using the same technologies as used in this study, had an inverse association with the development of Mets over time [19]. Additionally, relative skeletal muscle, such as the skeletal muscle to visceral fat ratio (SVR) measured through direct segmental in standing position at multifrequency BIA device, the same technologies as our study, was found to be negatively associated with an increased risk for exacerbating Mets [20,34].
And authors described this point as limitations in 4. Discussion (page 8, Line 42~45) as follows: And the body composition parameters, such as LBM, are dependent on technologies of measuring instrument. Therefore, it is necessary to be careful to compare with the results from other studies that estimate body composition with different technologies.
Thank you very much!

Round 2
Reviewer 3 Report
Thank you for the changes made.